# IPEX Syndrome: Genetics and Treatment Options

**DOI:** 10.3390/genes12030323

**Published:** 2021-02-24

**Authors:** Iwona Ben-Skowronek

**Affiliations:** Department of Pediatric Endocrinology and Diabetology, Medical University, 20-093 Lublin, Poland; iwonabenskowronek@umlub.pl; Tel.: +48-667-985-924

**Keywords:** IPEX syndrome, autoimmunity, FOXP3 mutations, HSCT, genetic engineering

## Abstract

(1) Background: IPEX (immune dysregulation, polyendocrinopathy, enteropathy, X-linked) syndrome characterizes a complex autoimmune reaction beginning in the perinatal period, caused by a dysfunction of the transcription factor forkhead box P3 (FOXP3). (2) Objectives: Studies have shown the clinical, immunological, and molecular heterogeneity of patients with IPEX syndrome. The symptoms, treatment, and survival were closely connected to the genotype of the IPEX syndrome. Recognition of the kind of mutation is important for the diagnostics of IPEX syndrome in newborns and young infants, as well as in prenatal screening. The method of choice for treatment is hematopoietic stem cell transplantation and immunosuppressive therapy. In children, supportive therapy for refractory diarrhea is very important, as well as replacement therapy of diabetes mellitus type 1 (DMT1) and other endocrinopathies. In the future, genetic engineering methods may be of use in the successful treatment of IPEX syndrome. (3) Conclusions: The genetic defects determine a diagnostic approach and prognosis, making the knowledge of the genetics of IPEX syndrome fundamental to introducing novel treatment methods.

## 1. Introduction

Immune dysregulation, polyendocrinopathy, enteropathy, X-linked (IPEX) syndrome characterizes complex autoimmune reactions beginning in the perinatal period and caused by dysfunction of the transcription factor forkhead box P3 (FOXP3) [1,2,3,4]. It is a rare hemizygous disorder that may occur perinatally or later in life, but it is most common in the first year of life. FOXP3 plays a key role in controlling the regulatory T cell subset, and its dysfunction leads to autoimmunity [1,2,3,4,5]. IPEX syndrome is inherited in boys in an X-linked recessive manner. The *FOXP3* gene is located in the centromeric region of X chromosome Xp11.23–Xq13.3. Mutations of this gene cause the deformation and malfunction of protein localized in the DNA-binding domain (forkhead domain). The abnormal protein cannot bind to its binding spot on the DNA, and in this situation, the development and function of T regulatory cells are impaired, with an observed loss of CD4 + CD25 + T regulatory cells and uncontrolled proliferation of activated CD4 + effector cells [6]. The immune system attacks its own tissues and organs as a consequence of uncontrolled T cell activation, resulting in inflammation, autoimmunity, and metabolic disorders [7,8,9,10,11]. IPEX syndrome is one of the autoimmune polyendocrine syndromes that lead to the dysfunction of multiple endocrine glands and multiorgan inflammation. The most common abnormalities are diabetes mellitus in the first month of life, extremely active atopic dermatitis (eczema), autoimmune thyroiditis, refractory diarrhea, hemolytic anemia, thrombocytopenia, neutropenia, and nephropathy [12]. A study performed by Gambinieri et al. showed the clinical, immunological, and molecular heterogeneity of patients with IPEX syndrome [6]. Only half of the examined children presented with a FOXP3 mutation, while the rest were diagnosed with an IPEX-like syndrome because of no identifiable FOXP3 mutations. Those patients presented an X-like syndrome connected to other phenotype dysregulation genes: *STAT5b*, *STAT1*, *STAT3*, *IL2RA*, *CTLA4*, *LRBA*, *TTC7A*, *TTC37*, *LRBA*, and *DOCK8* [6].

An overview of the current knowledge about the pathophysiology, genetics, and therapeutic perspectives is very important for the diagnosis and treatment of patients with IPEX syndrome and for understanding the mechanisms of autoimmunity.

## 2. Genetics of IPEX Syndrome

Forkhead box P3 (FOXP3), also known as scurfin, is from a family of proteins involved in immune system responses [13]. Scurfin is a member of the FOX protein family, the forkhead/winged-helix family of transcriptional regulators. FOXP3 presumably acts via similar DNA-binding interactions during transcription and is a pivotal regulator of the regulatory pathway in the development and function of regulatory T cells [1,2,3,13,14]. A deficiency in regulatory T cell activity in autoimmune diseases can allow other autoimmune cells to attack the body’s own tissues [15,16]. In regulatory T cell model systems, the FOXP3 transcription factor plays the role of promotor for genes engaged in regulatory T cell functions and may stop the transcription of key genes following the stimulation of T cell receptors [17]. Many transcription factors are shown to associate on these enhancers to regulate FOXP3 expression, one of those factors is runt-related transcription families (RUNX) and erythroblast transformation-specific (ETS) proteins.

FOXP3 is located in the short arm zone (Xp11.23) of the X chromosome [18]. FOXP3 consists of 12 exons in humans in a centromeric-to-telomeric orientation [12]. The full-length open-reading frame encodes a 431-amino acid (aa) protein. FOXP3 protein contains four domains: The N-terminal proline-rich region (PRR), the central zinc finger (ZF; aa 199–222), leucine zipper (LZ; aa 239–260), and the forkhead (FKH) domain (aa 335–418) [12,19]. The PPR, ZF, and LZ are involved in FOXP3 interactions. The FKH domain is necessary for DNA-binding activity [12,20]. The PPR is the repressor domain (aa 67–132) that mediates transcriptional activity by suppression of the nuclear factor of activated T cells (NFAT) [21]. FKH domain mutations are the most common in patients with IPEX syndrome [6]. Binding FOXP3 with sequence GTAAACA motifs by the FKH domain leads to immunosuppression, because it initiates the transcription of promoted genes (CD25, CTLA-4, GITR, CD103, and TNFRSF18) and suppresses genes (interleukin, IL2, Interferon-γ, IL4, and PTPN22) [12,13,22,23]. FOXP1 assists FOXP3’s binding to DNA through heterodimerization. The deletion of FOXP1 causes a reduction in IL-2 action [23]. The interaction of FOXP3 and retinoic acid-related orphan receptor γ (RORγt) can repress (RORγt)in this way, and transforming growth factor β (TGF-β) impairs TH17 cell differentiation [24]. The stability of transcriptional factor FOXP3 determines the character of regulatory T (Treg) cells. The epigenetic modifications, including DNA methylation and histone, have been described at the FOXP3 locus [21,25,26,27].

A novel element involved in the regulation of FOXP3 is noncoding RNAs. MicroRNAs (miRNAs) are responsible for RNA silencing and post-transcriptional regulation. The proteins associated with the processing of miRNAs inhibit FOXP3 and remove suppressor capacity [28,29]. These proteins are knocked out in Treg cells [12].

Phosphorylation, ubiquitination, acetylation, methylation, and poly(ADP (adenosine-5′-diphosphate)-ribosyl)ation have been identified as pivotal post-translational regulation mechanisms [30,31,32]. The new therapeutic methods to control inappropriate immune responses in IPEX syndrome are based on different enzymes.

In the literature, over 70 FOXP3 mutations associated with IPEX syndrome have been reported, but the relationship between the genotype and phenotype of IPEX syndrome needs explanation. Similar genotypes can result in different phenotypes—severe or mild forms of IPEX syndrome have been observed in children from the same family. The signs and symptoms of IPEX syndrome are related to the genetic changes in FOXP3: nonsense variants, missense variants, small in-frame amino acid deletions or insertions, and splice site variants [12,33,34,35].

Half of the patients with typical clinical symptoms of IPEX syndrome present with no FOXP3 mutation, but have other gene mutations. These children are diagnosed with IPEX-like syndrome, in which the genetic effect is unknown or other mutations are diagnosed, such as IL2RA, STAT 5b, ITCH, STAT1, STAT3, CTLA4, LRBA, TTC7A, and TTC37 [6,34,35,36,37,38,39,40,41,42]. These genes are related to Treg functions, as a classical clinical triad of symptoms; i.e., enteropathy, eczema, and endocrine disorders were observed in patients with IPEX-like syndrome [6].

The most common connections between the mutations and their effects on patients’ symptoms are described in Table 1. Recognition of the kind of mutation is important for the diagnostics of IPEX syndrome in newborns and young infants, as well as in prenatal screening. As shown by Gambineri and coworkers, stratification of IPEX patients for the mutation type is possible. The genotype may be correlated with signs and symptoms and with the severity of immune dysregulation [6]. The complete triad was mainly observed in patients affected by in-frame deletions and frameshift mutations [6]. Early detection of mutations connected with the severe IPEX syndrome form may contribute to intensification of therapy, earlier decision about hematopoietic stem cell transplantation (HSCT), and prediction of survival. The milder form of IPEX syndrome needs diabetes-type 1 insulin therapy or psoriasiform dermatitis treatment—HSCT in these patients may be considered at an older age.

**Table 1 genes-12-00323-t001:** The connection between the mutations and clinical symptoms in immune dysregulation, polyendocrinopathy, enteropathy, X-linked (IPEX) syndrome.

Mutation.	Effect	Patients’ Symptoms
**c.1150G >****A** [43]	Missing FKH domain	Fetal onset and severe nonviable neonatal patients survived to adolescence
**c.1189C >****T** [6,44]	Located in the FKH domain	Fetal onset and severe nonviable neonates
**c.384A >****T mutation** [22,45]	Lying within a helix H3 can influence DNA contact	Severe form of IPEX syndrome
**c.319-32delTC** [16]	Deleting two-base pair in the N-terminal domain	Neonatal death
**c.227delT mutation** [46]	Frameshifts and premature stop codons	
**c.303_304delTT in exon 3** [46]		
**p.Glu251 del in LZ-FKH loop** [47]	The LZ region is important for forkhead box P3 (FOXP3) to form dimers that impairs FOXP3′s capability to inhibit IL-2 transcription	
**p. Glu251 del**	Influences the inter-subunit salt bridge, leads to a disruption of FOXP3 homodimerization and an impaired inhibiting ability	
**p.Lys250del** [48]		
**F371C, F373A, and p.R347H** [49,50]	The ability of FOXP3 to inhibit transcription can be impaired	
**Mutations****occur in the promoter and****5′****untranslated region of FOXP3** [6]		Milder form of IPEX syndrome
**g.6247–4859del** [51]	Encompassing the upstream noncoding exon and the adjacent intron of FOXP3 accumulation of unsliced pre-mRNA and alternatively spliced mRNA	Enteropathy, impressive allergic phenotype
**p.Arg114Trp, p.Arg347His, p.Lys393Met, and c.1044 +****5G >****A** [52]		Insulin-dependent diabetes without other characteristics of IPEX syndrome
**c.1150G >****A** [53,54]	Changes in the DNA-binding site	Psoriasiform dermatitis and alopecia universalis
**Alternative** **splicing of FOXP3**	Exon 3 encoding blocking retinoic acid-related orphan receptors α and γt, exon 8 encoding a LZ motif necessary to FOXP3 function	Milder IPEX phenotype
**exon3 as exon 2** [55]		
**exon 8 as exon 7** [19]		

The occurrence of the most common mutations is shown in Figure 1 [56].

## 3. Epidemiology

The occurrence of IPEX syndrome is below 1:1,000,000.

IPEX syndrome is a result of the X-linked recessive disorder of FOXP3 and usually affects boys. The diagnosis of IPEX syndrome is connected to several typical clinical symptoms. The confirmation of diagnosis needs the identification of a hemizygous pathogenic variant in FOXP3. In boys, FOXP3 allele profiles are observed in peripheral blood mononuclear cells (PBMCs), including naïve and conventional T (Tconv) cells; however, natural Treg cells express active wild-type (WT) FOXP3 [53]. In females, the mutation must occur in both copies of the gene to cause the immune disorder, and this is an extremely rare situation.

During pregnancy, the development of fetal tissue needs maternal tolerance. The occurrence of IPEX syndrome is associated with recurrent male miscarriage and fetal akinesia [54].

## 4. Clinical Features and Their Pathophysiology

IPEX syndrome is diagnosed in the first year of life in males as multiple autoimmune diseases syndrome. In some patients, a milder form of the disease appears after the first year as late-onset IPEX syndrome [55]. IPEX syndrome is characterized by three main clinical manifestations: enteropathy, endocrinopathy, and dermatitis. Park et al., after a systemic review of 75 articles and data from 195 patients, presented the occurrence of the clinical manifestations of IPEX syndrome: enteropathy (97.7%), skin manifestations (62.1%), endocrinopathy (53.3%), hematologic abnormalities (38.5%), infections (40%), other immune-related complications (22.1%), and renal diseases (16.4%) [56].

### 4.1. Enteropathy

The first symptom of IPEX syndrome is usually intractable autoimmune enteropathy. The infiltration of lamina propria and mucosa with activated T cells leads to villous atrophy, with B cells secreting large amounts of IgA and IgE, as well as producing antiharmonin autoantibodies (HAAs) and antivillin autoantibodies (VAAs) [57]. HAAs and VAAs may be used as specific markers of IPEX syndrome enteropathy. Damage to the intestines leads to malabsorption syndrome, maldevelopment, and weight loss; thus, children need parenteral nutrition [6]. In some cases, intractable diarrhea may develop during the first six months of life. Villin and harmonin are distributed in proximal renal tubules as well, and increased levels of HAAs and VAAs have been observed, as has nephropathy [43].

### 4.2. Endocrinopathy

Endocrinopathy occurs in the first year of life, and diabetes mellitus type 1 (DMT1) or autoimmune thyroid disease (AITD) may develop, usually in the first month of life. These autoimmune diseases may develop as the result of extreme autoimmunological reactions of activated T cells [14,43,58]. DMT1 treatment is extremely difficult due to intractable diarrhea and problems with food intake. Treatment with individual pump therapy and continuous glucose monitoring is difficult due to skin eczema. AITD can occur as autoimmune thyroiditis with hypothyroidism or thyrotoxicosis.

### 4.3. Dermatitis

The most common form of dermatitis is eczema, with abnormal red patches. Psoriasiform or ichthyosiform dermatitis may be observed as well. Elevated levels of IgE in the skin may result in skin desquamation on the limbs, bullae, urticaria, and alopecia universalis [44,59].

### 4.4. Hematologic Disorders

Patients with IPEX syndrome may present autoimmune bone marrow diseases such as hemolytic anemia, neutropenia, and thrombocytopenia. In children, splenomegaly and lymphadenopathy induced by autoimmune lymphoproliferation is observed [14,43].

### 4.5. Other Autoimmune Diseases

Children with IPEX syndrome may develop autoimmune hepatitis and nephropathy. In this situation, children are predisposed to developing invasive diseases: sepsis, meningitis, pneumonia, and osteomyelitis [43]. The outcomes of these patients are usually poor, potentially resulting in death in no less than two years. Children with IPEX syndrome usually present with the classical triad of eczema, DMT1, and enteropathy, but IPEX-like syndrome patients are frequently affected by autoimmune thyroid disorders and cardiovascular abnormalities [6]. The number of diseases in IPEX syndrome may increase with age [6,12,43].

The occurrence of IPEX clinical features is shown in Figure 2.

## 5. Diagnostic Methods

IPEX syndrome is a disease of immune dysregulation [60]. The diagnosis of IPEX syndrome is based on characteristic clinical symptoms in male patients. The combination of eczema and diarrhea as the result of enteropathy needs differentiating from carbohydrate malabsorption and food allergies, viral and bacterial infections, cystic fibrosis, congenital enterocyte defects, and abetalipoproteinemia.

In these patients, laboratory tests are performed: estimation of IgA, IgG, IgM, complement level, and IgE (in eczema). Eosinophilia and elevated IgE level are typical. Estimation of HAAs and VAAs is helpful. Decreased Treg lymphocyte subsets may be observed in flow cytometry, as the FOXP3 expression may be reduced. Treg cells are usually present [6]. In half of the patients, the T cell number, B lymphocyte function, polymorphonuclear leukocyte chemotaxis, and complement concentrations are normal. Genetic tests are pivotal [6]. Differential diagnosis is very important with other causes of neonatal diabetes (genetic tests). To confirm autoimmune processes glutamic acid decarboxylase (GAD) antibodies, IA2 antibodies, ZnT8 antibodies in children with DMT1 and thyroperoxodase (TPO) Ab, and thyroglobulin (TG) Ab in children with autoimmune thyroiditis may be estimated. Genetic tests for FOXP3 gene mutation should be performed in children with a triad of symptoms and elevated IgE, especially in patients with a decreased number of Treg cells. Neonatal diabetes is an important indication for genetic tests as well. Hwang et al. have shown that some FOXP3 variants result only in insulin-dependent diabetes such as p.Lys393Met and c.1044_5G > A [52]. A differential diagnosis between symptoms of IPEX syndrome and many other inherited immunological disorders is difficult. In this situation, exome sequencing and genome sequencing are recommended to confirm disease types.

The prenatal ultrasound findings of echogenic bowel and skin desquamation are associated with a diagnosis of IPEX syndrome. This broadens our knowledge about the phenotypic spectrum of IPEX syndrome and proves the feasibility of prenatal diagnosis [33]. This has especially been observed in patients with IPEX syndrome and antibodies against HAAs and VAAs [57].

## 6. Treatment for IPEX Syndrome

The treatment of IPEX syndrome has three directions: downscale of autoimmunity, reduction of inflammation due to autoimmunity disorders, and treatment of infections as complications of the destroyed immunity and therapeutic methods. Currently, the primary therapeutic approach for IPEX syndrome is allogeneic hematopoietic stem cell transplantation (HSCT). Patients without HSCT are considered for immunosuppressive therapy (IS) and long-term supportive care.

Before diagnosis, patients usually require supportive care in hospital. For patients with severe enteropathy, nutritional treatment, including total parenteral nutrition or elemental or low-carbohydrate-containing formula and fluids, is indicated. Patients with neonatal diabetes type 1 need insulin therapy and continuous glycemic control. L-thyroxin is necessary for patients with hypothyroidism due to AITD. Other replacement therapies with glucosteroids and platelet transfusion are used in thrombocytopenia. Autoimmune granulocytopenia and pancytopenia need special hematologic procedures. Eczema is treated with antiallergic ointments, balms, and steroids.

Human stem cells have the competence to differentiate into mature blood cells including Treg cells. Allogeneic HSCT is the therapy of choice for IPEX syndrome. Cells from either human leukocyte antigen (HLA)-identical siblings or other HLA-matched donors are used for HSCT [61]. Patients receive HSCT from cord blood, peripheral blood, or bone marrow [62,63]. The survival of patients after HSCT is strongly connected with patients’ manifestation before transplantation. The severity of a patient’s disease may cause a high death rate in the early years after transplantation. In a long-term follow-up conducted by Barzaghi et al., which evaluated 58 patients who received HSCT, the overall survival after HSCT was 73.2% [43]. In this study, the mortality after HSCT had poor correlation with the stem cell source, type of donor, and chimerism [43]. Only partial donor chimerism was sufficient for complete IPEX syndrome remission [64,65]. Immunosuppressive therapies are used with a combination of several immunosuppressive agents to control the acute autoimmune phase of IPEX syndrome. Patients after transplantation also receive immunosuppressive therapy, usually consisting of calcineurin inhibitors, cyclosporine A, sirolimus and tacrolimus, while some patients also receive steroids [66]. The current study considered rapamycin, a non-calcineurin inhibitor, as the primary choice of immunosuppressive drug, either used alone or in combination [43,65]. According to Barzaghi et al., the overall survival after immunosuppressive therapies is 65.1% [43]. Immunosuppressive therapies without transplantation do not change the disease progression, and the organ involvement score is decreased in long-term follow-ups, which has a negative impact on survival. In patients receiving chronic immunosuppression, disease recurrence or complications have been observed. HSCT results in disease resolution with a better quality of life, regardless of age, donor source, or conditioning regimen [43].

Long-term survival (30 years) has been indicated in 47.5% of patients with IPEX syndrome; however, it is higher in the IPEX group (52%) in comparison to the IPEX-like group (27%). The stratification of types of mutation in children with a 10-year survival can detect that the highest 10-year survival (87.5%) is connected with frame deletion and that 78% is connected with splice site mutations. The effect of HSCT according to 10 year follow-ups is the survival of 72.8% of patients with IPEX syndrome and 100% of patients with IPEX-like syndrome. Non-transplanted patients with IPEX syndrome survive in 57% of cases, but those with IPEX-like syndrome survive in 78% [6].

### New Treatment Perspectives

The new treatment methods for IPEX syndrome are based on genetically reprogramed techniques. Genetic engineering allows the use of Treg-like cells from patients’ autologous cells after replacing or repairing the FOXP3 mutants with wild-type genes [67,68,69,70].

Functional human Treg cells can be developed by a lentivirus FOXP3 transfer to naïve and memory CD4 + T cells [69,70,71]. Nowadays, the development of gene therapies for IPEX syndrome by generating CD4 + T cells expressing wild-type FOXP3 is used. T-like cells CD4FOXP3 have potent suppressor functions, both in vitro and in the humanized murine model [67,68,69]. Successes in HSCT treatment, especially with genetically changed wild-type FOXP3-expressing hematopoietic stem cells, could be considered, but there is also a problem of FOXP3 overexpression, because it has a negative influence on hematopoietic precursors [72,73,74,75,76,77].

Genome editing uses a site-specific endonuclease to produce a DNA double-strand break (DSB), and sequences in the DSB can be repaired by a donor sequence, or nonhomologous end joining [12,78]. Genome editing is a promising method in the murine model and will probably have a better long-term therapeutic effect in comparison to gene transfer [78,79,80,81].

An innovative method of genome editing, i.e., clustered regularly interspaced short palindromic repeats (CRISPRs) and CRISPR-associated (Cas) systems (CRISPR/Cas9 RGENs), is simple and has shown potent activity in a range of different cell types [82]. In this method, a wild-type FOXP3 complementary DNA (cDNA) is precisely inserted. Wild-type cDNA is expressed and regulated from the endogenous regulatory elements. The correct design of the donor construct is very important as well, and any splicing regulation can also be maintained. Finally, wild-type cDNA restitutes the function of endogenous genes. Current investigations are ongoing to test the advantage and feasibility of gene correction for IPEX syndrome [81,82]

Treatment with mTOR inhibitors, which can restore Treg cell function in IPEX syndrome, with decreased cytokine expression and elevated suppression ability is also a promising method [82].

## 7. Concluding Remarks

IPEX syndrome is clinically, immunologically, and molecularly heterogenic. Only half of the examined children presented FOXP3 mutations, while the rest were diagnosed with IPEX-like syndrome. For confirmation of the disease type, genomic testing (exome sequencing and genome sequencing) is recommended, because the symptoms, treatment, and survival are closely associated with the genotype of IPEX syndrome. Recognition of the kind of mutation is important for the diagnostics of IPEX syndrome in newborns and young infants, as well as in prenatal screening. The method of choice in the treatment of IPEX syndrome is HSCT and immunosuppressive therapy. In children, it is very important to include supportive therapy for refractory diarrhea and replacement therapy in DMT1 and other endocrinopathies. In the future, genetic engineering methods may be used in the successful treatment of IPEX syndrome.

## Figures and Tables

**Figure 1 genes-12-00323-f001:**
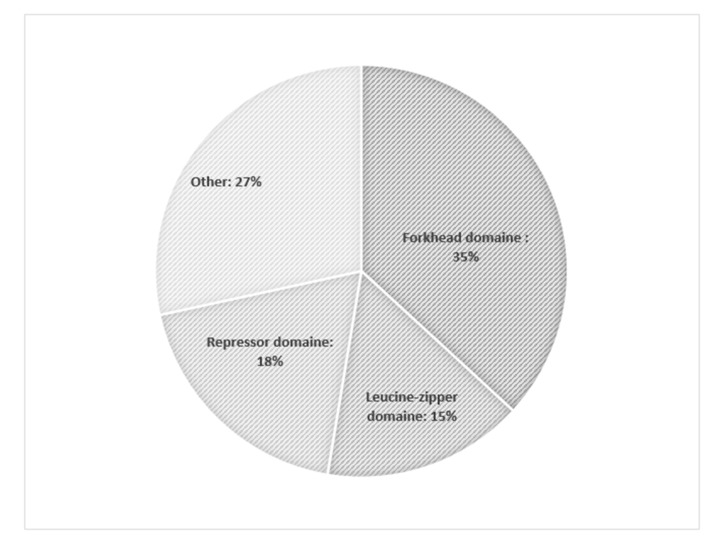
The occurrence of most the common mutations of forkhead box P3 (FOXP3) in patients with IPEX (immune dysregulation, polyendocrinopathy, enteropathy, X-linked) syndrome according Park et al. [56].

**Figure 2 genes-12-00323-f002:**
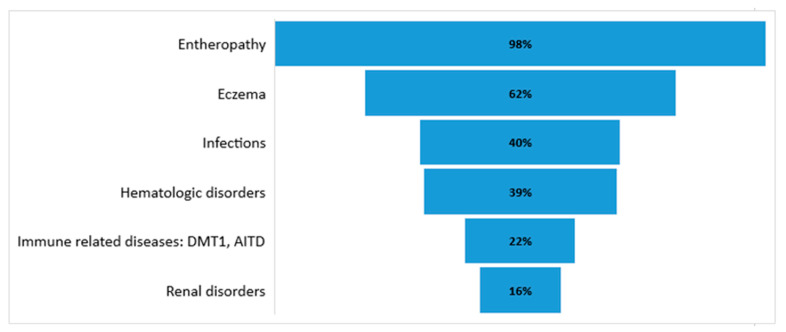
The occurrence of clinical symptoms of IPEX syndrome [56].

## Data Availability

Publicly available datasets were analyzed in this study.

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
