# Peer review of "IPEX Syndrome: Genetics and Treatment Options"

_genes, 2021, doi:10.3390/genes12030323_

Round 1

Reviewer 1 Report

The review is globally interesting and comprehensive, but an extensive English language editing is needed. Text plagiarim correction is also needed.

Author Response

Dear Sir/Madam

Thank you very much for your valuable advice. The English language has been improved and plagiarism eliminated by the Genes English service.

Reviewer 2 Report

The focus of this review is an update on IPEX syndrome. It is as I wrote an interesting subject and wrote while reviewing. Most of the data presented is not novel. Unfortunately the paper is very hard to read and follow, sentences are long and awkward, syntax is poor, and overall at its current state not meeting a level worthy of publication.

English language editing is mandatory , thre are numerous grammer, spelling mitakes. Overall revision and editng will help claryfing the ideas and informtion.

The author does address the subject. The text is in a great need for a thorough language editing, organization of the structure , and improvement in the flow of the entire manuscript.

Author Response

Dear Sir/Madam

Thank you very much for your valuable remarks.

  1. I have corrected the abstract according to your remarks.
  2. The English language has been proofread by the Genes English service.

Reviewer 3 Report

The review is about IPEX syndrome, a rare genetic disorder leading to severe immune dysregulation.

Unfortunately there are several shortcomings.

  1. The abstract does not clearly formulate the objective of the study. Also the conclusion :"The genetic defects determine a diagnostic approach and prognosis," does not make sense for me
  2. what is "phenotype dysregulation gene"?
  3. Especially the section "Genetics of IPEX Syndrome"  suffer from poor stucture: it is a collection of random facts. Lots of information here should be rewritten and collected under a more appropriate subheading.
  4. Lines 29-30 I don't understand: " Mutation of this gene caused by deformation and malfunction of protein localized in the DNA-binding domain (forkhead domain)." The mutations have been found in all domains of FOXP3.
  5. The first studies linking Foxp3 to treg function were not cited (Fotentot, Khattri, Hori)
  6. Please see Ref. 19 for FOXP3 binding motives: Ets family and Runx, not so much GTAAACA.
  7. What is the message from Table 1. Needs elaboration in the text.
  8. Diagnostic methods are kind of misleading. Do you start with autoantibody measurement in this case? Increased IgE and eosinophilia could be more helpful but definitely genetic testing that is guided by the severe phenotype is most important. Not sure if intestinal biopsy would be necessary. Moreover, the author has not described what to expect and how to differentiate from other causes of diarrhea. Not all the patients have decreased numbers of Tregs so the suggestion that genetic testing should be undertaken only in pts with decreased numbers of Tregs is misleading. 
  9. CRISP/Cas9 classifies also as method of genome editing
  10. Concluding remarks are partially not about the current study but about Ref 2.

Author Response

Dear Sir/Madam

Thank you very much for your valuable remarks.

  1. The English language has been proofread by the Genes English service.
  2. The phenotype dysregulation gene is according to the 5th Conference on Bioinformatics, Computational Biology and Health Informatics : one of the dysregulated phenotype-related interacting genes
  3. The section Genetics of IPEX Syndrome has been rewritten according to your advice.
  4. In lines 29-30, the following sentence has been corrected: Mutations of this gene localized in all domains cause deformation and malfunction of protein FOXP3.
  5. The studies linking FoxP3 to Treg function have been cited as positions 1, 2, 3, and 4 in the References.
  6. The FOXP3 binding motif has been described as the Ets family, Runx, and GTAAACA according to your suggestion.
  7. The description of table 1 in the text has been rewritten.
  8. The diagnostic methods have been improved according to your suggestion.
  9. CRISP/Cas9 has been classified to genome editing methods.
  10. The concluding remarks have been corrected.

Round 2

Reviewer 1 Report

The manuscript has been significantly improved.

Author Response

Dear Sir/Madam

Thank you for your comment that the manuscript has been significantly improved.

Reviewer 2 Report

Overall the manuscript is better and much more readable than previous version. However, there are some aspects that are missing. Such an important one is the lack of an insightful and engaging discussion.

As an example - the authors attempts to tackle the subject of genetics and genotype - phenotype, correlation, alas, I could not understand clearly if there is any, and I could not see any substantial discussion on the subject.

The author names a few IPEX-like diseases. Some of these are extremely different, and I am not sure that easily confused with IPEX (eg. STAT3, DOCK8). Such a claim should be supported an discussed in a little more detail.

Another significant problem is the flow and coherence of the text. In quite a few of the paragraphs, sentences are not connecting to well. Below you will find two examples:

In the diagnosis section- the marked sentence does not seem to be connected nor does it stems from previous ones.

"Genetic tests for FOXP3 gene mutation should be performed, especially in patients with a de-creased number of Treg cells. In children with intractable diarrhoea, a biopsy of the intes-tinal mucosa is important, and differential diagnosis with other causes of diarrhoea. Ele-vated IgE is significantly higher in children with IPEX syndrome in comparison to patients with IPEX-like syndrome [6]. Hwang et al. 2].showed that some FOXP3 variants result only in insulin-dependent diabetes such as p.Lys393Met and c.1044_5G > A [52]."

In the treatment section- this sentence is very unclear.

"The treatment of IPEX syndrome has three directions: autoimmunity Autoimmunity, inflammation due to autoimmunity disorders, and infections as complications of the de-stroyed immunity and therapeutic methods."

Author Response

Dear Sir/Madam

Thank you for your  valuable remarks. According to your sugestions  I have expanded the discussion about table 1 and about IPEX-like diseases. I have improved  the flow and coherence in the text especially in the diagnosis and the treatment sections.